# Vascular Flora on Croatian Historic Structures: Drivers of Biodeterioration and Conservation Implications

**DOI:** 10.3390/plants14121773

**Published:** 2025-06-10

**Authors:** Alessia Cozzolino, Riccardo Motti, Ivana Vitasović-Kosić

**Affiliations:** 1Department of Agricultural Sciences, University of Naples Federico II Via Università, 100, 80055 Portici, Italy; alessia.cozzolino2@unina.it (A.C.); motti@unina.it (R.M.); 2Division of Horticulture and Landscape Architecture, Department of Agricultural Botany, University of Zagreb Faculty of Agriculture, Svetošimunska cesta 25, 10000 Zagreb, Croatia

**Keywords:** invasive species, Raunkiær life forms, cultural heritage conservation, vascular plants, hazard index

## Abstract

Biodeterioration, the alteration of materials by living organisms, affects approximately two-thirds of the world’s cultural heritage. When organisms colonize the surfaces, they induce physical and chemical changes that can lead to significant damage. Despite its relevance, this phenomenon remains understudied in Croatia. This study aims to assess the deteriogenic vascular flora colonizing Croatian historical structures, including castles, towers, and archaeological remains, in relation to several environmental and anthropogenic factors: distance from the sea (0–1, 1–10, 10–65, and 65–165 km), elevation (0–50, 50–150, 150–300, and 300–600 m a.s.l.), exposure (north, south, east, west), and the state of conservation of the site (absent, low, good, excellent). Vegetation cover and floristic diversity, assessed using the Shannon Index, were primarily influenced by elevation and conservation status. As expected, vegetation cover decreased significantly, by 67.75%, from sites classified as ‘Absent’ to those with ‘Excellent’ conservation status (*p* < 0.001). To explain the observed differences in vegetation cover across the four altitudinal ranges, an analysis of plant life forms was carried out, revealing a wide variability and statistically significant patterns also related to the type and frequency of maintenance interventions. The potential risk posed by vascular plants was evaluated using the Hazard Index (HI), which revealed significant differences only for elevation and distance from the sea. The highest risk levels were recorded at mid-elevations (150–300 m), where the Hazard Index reached its maximum value (HI = 158). Exposure did not show a significant effect on biodeterioration processes. These findings provide new insights into plant-driven biodeterioration in Croatia and highlight the need for targeted conservation strategies to protect the country’s cultural heritage.

## 1. Introduction

Stone monuments and historic buildings are essential components of World Cultural Heritage across all continents [1]. Heritage plays a crucial role in fostering community cohesion and identity, serving as a fundamental pillar for social and community harmony [2]. However, the artefacts, buildings, and sites that constitute this heritage are constantly exposed to abiotic deterioration factors such as wind, rain, sunlight, and humidity [3]. These factors are further compounded by biological agents, including bacteria, lichens, algae, fungi, bryophytes, and vascular plants that cause several types of damage [4]. These organisms interact with historic materials, such as stone and other construction elements, by assimilating or transforming environmental compounds [5]. Consequently, the resulting alterations can be both chemical and physical. Biochemical damage occurs through direct metabolic activity with production of acids, chelating substances, alkalis, enzymes, and pigments. In contrast, biophysical damage is associated with mechanical degradation processes, such as the penetration and expansion of organisms within the material, or through cyclical hydration and dehydration [6]. The complexity of biodeterioration phenomena observed on cultural heritage materials is influenced by multiple factors, including the chemical composition and intrinsic properties of the substrate, climatic conditions, exposure, and the methods and frequency of surface cleaning [7]. According to Guillitte’s concept, the susceptibility of a substrate to colonization by living organisms is referred to as bioreceptivity [8,9]. Bioreceptivity is a dynamic property that evolves over time, leading to different forms. Initially, a material possesses primary or intrinsic bioreceptivity, which depends on its inherent characteristics [10]. However, once exposed to environmental conditions and weathering processes, its bioreceptivity changes, giving rise to secondary bioreceptivity. Maintenance interventions, such as the application of protective coatings, introduce tertiary bioreceptivity, while the accumulation of external materials, such as dust, contributes to extrinsic bioreceptivity [11]. Among the various biological agents, vascular plants pose one of the most significant threats to the integrity of monuments and historic structures. This is due to their production of acid metabolites, their ability to penetrate building materials with their roots, and their growth in the spaces between stones [12,13]. The colonization process typically begins with pioneer species, such as herbaceous annuals and perennials, which cause minimal damage [14,15]. Over time, these are replaced by more destructive plants, including small shrubs and trees. The vegetation growing on buildings and walls generally originates from the surrounding environment, where propagules disperse and establish themselves on exposed surfaces [12]. Plant colonization occurs through various processes. In some cases, plants take root in cracks, where accumulated dust and atmospheric particles form a thin layer of soil, creating a suitable substrate for growth. Moreover, droppings from birds and other small animals contribute to this process by enriching the substrate with organic matter and introducing seeds from the surrounding vegetation, further facilitating plant establishment. Alternatively, seeds may germinate within a moss layer, which serves as a natural substrate, providing a humid and protective environment that supports plant development [16]. Given the significance of this issue, research in recent decades has increasingly focused on higher plants [17,18,19,20,21,22,23,24]. This study aims to assess the deteriogenic vascular flora colonizing historic buildings in Croatia, including castles, towers, and archaeological remains, while also evaluating their potential impact on structural integrity. All surveyed structures are built exclusively with limestone, a material whose physical and chemical properties influence its bioreceptivity and response to plant colonization. Croatia has long been a prominent international tourist destination, particularly since the expansion of large-scale tourism.

Despite the exceptional value of its cultural heritage, only a limited number of historic and cultural sites are actively integrated into local tourism offerings [25]. In neighboring countries such as Italy and Greece, numerous historic and archaeological sites have been studied for biodeterioration, yielding important insights and contributing to their scientific and cultural valorization [15,20,24,26]. In contrast, this phenomenon remains poorly investigated in Croatian monuments. A comprehensive study in this context could offer valuable contributions to the understanding of Croatia’s cultural heritage, fostering an integrated perspective between botany and history. In particular, the analysis revealed the presence of both endemic and invasive vascular plant species, highlighting the need to consider floristic composition not only from a conservation biology perspective but also in relation to management priorities.

## 2. Results

### 2.1. Floristic Surveys

A total of 154 taxa from 58 families were documented across the 40 study sites (Appendix A, Appendix A). Figure 1 shows the cumulative increase in species richness as the number of collected samples and surveyed sites increases.

The families with the highest species richness were Asteraceae (22 taxa), followed by Poaceae (12 taxa) and Lamiaceae (7 taxa). Among the recorded species, *Parietaria judaica* L. was the most widespread, occurring in 34 out of 40 study sites, with a total of 246 occurrences. *Hedera helix* L. followed, being present in 24 sites with 77 occurrences, while *Ficus carica* L. and *Asplenium ceterach* L. were recorded in 21 and 20 sites, with 44 and 55 occurrences, respectively (Appendix A, Appendix A). From a chorological perspective, several alien taxa were identified during the surveys, including *Ailanthus altissima* (Mill.) Swingle, *Robinia pseudoacacia* L., *Parthenocissus quinquefolia* (L.) Planch., *Erigeron sumatrensis* Retz., *Erigeron annuus* (L.) Desf., *Euphorbia prostrata* Aiton, and *Euphorbia maculata* L. In contrast, the endemic taxa recorded were *Campanula fenestrellata* subsp. *fenestrellata*, *Campanula fenestrellata* subsp. *istriaca* (Feer) Damboldt, *Centaurea dalmatica* A. Kern., and *Centaurea spinosociliata* Seenus.

Figure 2 presents the average species coverage across the study sites, with Fort Munida in Pula showing the highest species abundance (82.5%), followed by Zelingrad Castle in Kladešćica (82%) and Salatić Castle in Vrh (79.5%).

The Bray–Curtis dendrogram (Figure 3) illustrates the clustering of study sites based on species composition, ordered according to the Jaccard Index of Association. The heatmap reveals several distinct clusters: (1) a cluster including Zelingrad, Novigrad, and Medvedgrad Castles; (2) a cluster encompassing the fortresses of Munida, Valmaggiore, Fort Bourguignon, Giorgio, Grosso, Punta Cristo, Casoni Vecchi, San Michele, and the church of Mirine; (3) a cluster comprising Andautonia, the walls of Krk, the Amphitheatre of Pula, the Roman Temple of Nin, Kličevica, as well as the castles of Krk, Morosini, Pula, Benković, Brinje, and Budak tower.

One-way ANOVA tests revealed significant differences among the three floristic clusters identified in the heatmap, both in terms of elevation (F(2, 19) = 7.178, *p* = 0.005) and distance from the sea (F(2, 19) = 16.691, *p* < 0.001). Post hoc comparisons using Tukey’s HSD test showed that Cluster 1 included sites at significantly higher elevations (mean = 361 m a.s.l.) and greater distances from the sea (mean = 113,152 m) compared to Cluster 2 (mean elevation = 43 m, mean distance = 396 m), while Cluster 3 occupied intermediate positions in both variables and did not differ significantly from the other clusters.

### 2.2. Vegetation Response to Selected Variables

Figure 4 presents the analysis of average vegetation cover across the considered variables. The continuous variables were grouped into four ranges: distance from the sea (0–1, 1–10, 10–65, and 65–165 km) and elevation (0–50, 50–150, 150–300, and 300–600 m a.s.l.). Additionally, four exposition categories and four levels of conservation status were considered: absent, low, good, and excellent.

Distance from the sea and exposure do not show statistically significant differences, suggesting that these environmental factors are not primary drivers of vegetation cover (F(3, 386) = 0.416, *p* = 0.742; and F(3, 440) = 1.551, *p* = 0.201, respectively). In contrast, elevation and conservation status show significant differences among subgroups (F(3, 402) = 4.259, *p* = 0.006; and F(3, 448) = 37.261, *p* < 0.001, respectively). In Figure 5 there is the analysis of life form distribution across elevation and state of conservation ranges. For the elevation, phanerophytes, chamaephytes, and therophytes together account for 45% and 48% of total cover in the first two ranges, whereas their combined cover drops to 32% in the higher ranges.

Regarding conservation status, sites with better maintenance practices exhibit lower levels of colonizing vegetation, as reflected in the cover percentage values. Notably, phanerophytes are more abundant in sites with absent or poor maintenance, whereas hemicryptophytes, typically small and with limited spatial spread, tend to increase progressively along the conservation gradient.

Floristic diversity was also influenced by environmental and structural conditions, including elevation, distance from the sea, and conservation status, as shown in Figure 6. Diversity was assessed using the Shannon Index, which revealed no significant differences in relation to exposure (F(3, 440) = 0.164, *p* = 0.92), in contrast to the other variables considered: elevation (F(3, 448) = 14.679, *p* < 0.001), distance from the sea (F(3, 448) = 19.620, *p* < 0.001), and conservation status (F(3, 448) = 20.573, *p* < 0.001).

A significant reduction in the Shannon Index was observed at mid-elevations (150–300 m a.s.l.), whereas both lower (0–150 m) and higher elevations (>300 m) supported greater floristic diversity.

Along the coastal–inland gradient, sites located at intermediate distances from the sea (10–65 km) exhibited significantly lower floristic diversity compared to both coastal (0–1 km) and inland sites (65–165 km).

Finally, sites with low or absent conservation showed the highest levels of floristic diversity, while more well-maintained structures exhibited a significant reduction in diversity.

### 2.3. Hazard Index

To evaluate the potential impact of the deteriogenic flora observed at the study sites, the Hazard Index (HI) was calculated as the product of each species’ HI and its percentage cover, and subsequently analysed in relation to environmental and conservation variables (Figure 7). The analysis revealed statistically significant differences only across altitudinal and distance ranges (F(3, 448) = 6.678, *p* < 0.001; and F(3, 448) = 2.80, *p* = 0.039, respectively), while no significant effects were found for exposure and conservation status (F(3, 440) = 1.551, *p* = 0.201, and F(3, 448) = 0.975, *p* = 0.404, respectively).

Across the first three distance bands there are high values of HI, in contrast, the farthest range (65–165 km) recorded the lowest HI values. As for elevation, the highest HI values were observed in the 150–300 m a.s.l. range, while in the remaining elevation bands, particularly in the highest one (300–600 m a.s.l.), there is a decreasing of HI values.

## 3. Discussions

This study provides one of the first systematic evaluations of deteriogenic vascular flora affecting historical structures in Croatia, offering new insights into the environmental and anthropogenic factors shaping plant-driven biodeterioration.

Floristic distribution in this study was assessed in relation to environmental and management variables. In many studies, including that of Motti et al. [17], substrate is considered an additional factor influencing floristic composition, particularly when multiple substrate types are present, as this allows for correlation analyses between vegetation and lithological variability. In the present case, however, all sites were built exclusively on limestone; for this reason, substrate was not included among the variables considered. An analysis of average vegetation coverage across the 40 study sites (Figure 2) reveals substantial variability, largely influenced by geographic location and the extent of maintenance activities, often linked to their touristic relevance. Focusing on sites with over 50% vegetation coverage, these areas are typically located within densely wooded environments, where the absence of regular cleaning and maintenance fosters the proliferation of arboreal species. Notably, this includes the fortresses of Pula and the castles of Zelingrad, Salatić, and Duecastelli, where extensive growth of lianescent climbers such as *Hedera* and *Clematis* was also observed.

Among the most represented families, Asteraceae were particularly abundant, a pattern frequently observed and reported in previous studies [4,27]. This predominance is attributable to the ecological traits of the family, as many Asteraceae species are capable of thriving under unfavourable conditions due to their broad ecological tolerance and rapid seed germination and growth.

While the analysis of average vegetation coverage offers a general overview of site conditions, the heatmap in Figure 3 provides a more detailed perspective by highlighting floristic patterns and site-species affinities. Specifically, the heatmap revealed three distinct clusters of sites, each characterized by a different composition of dominant species.

In the first cluster, the three identified castles are located in the inland region of Croatia. These sites are characterized by the presence of different phanerophytes, such us *Robinia pseudacacia* L., *Fraxinus ornus* L., *Paliurus spina-christi* Mill., *Quercus pubescens* Willd., *Populus alba* L., *Populus nigra* L. and *Corylus avellana* L. The presence of many phanerophytes could be linked to the presence of very rich forested areas. Notably, the study sites belonging to this cluster are located in inland regions of Croatia (65–165 km), where natural woodland vegetation is more prevalent. Medvedgrad castle is situated on the south slopes of Medvednica mountain, while Zelingrad castle is on the northern slopes. The vegetation of this mountain is characterized by the presence of various forest types, including beech and sessile oak forests, as well as forests of sweet chestnut and black alder [28,29]. While *Populus alba* is not a predominant species in these formations, it may occur in transitional zones or wetter areas, particularly where the soil is deep and moist. Additionally, *Fraxinus ornus* and *Corylus avellana* are commonly found in sessile oak and chestnut forests, while *Populus nigra* is primarily restricted to riparian zones and particularly humid areas within black alder forests. In contrast, *Paliurus spina-christi* is not a characteristic species of these formations, yet it can thrive in a variety of bioclimatic and successional contexts [30]. Its presence in certain areas may result from seed dispersal by mammals or birds, contributing to its establishment outside its typical range. The Castle of Novigrad na Dobri is located on a hill overlooking the Dobra River in the Karlovac region of Croatia. The surrounding landscape is characteristic of the hilly areas of the Croatian hinterland, consisting of a mosaic of mixed forests and agricultural land. There is high presence of *Populus nigra*, a key species of riparian vegetation, and *Robinia pseudoacacia*, an invasive non-native tree. This invasive non-native tree is a pioneer species that rapidly colonizes poor, disturbed, or degraded environments, as well as forest edges, such as those found in the castle’s surroundings [31,32].

The second cluster includes sites located in the coastal area (0–1 km), particularly the Church of Mirine and, predominantly, the fortresses in Pula, constructed by the Austro-Hungarian Empire to protect the strategic naval port [33]. These structures were built primarily using local limestone, probably sourced from the Roman-era “Cavae Romanae” quarry near Pula, the same material used for the Pula Amphitheatre [34], and were later reinforced with concrete and steel to enhance their durability and defensive capabilities [35]. Even in this case, phanerophytes are abundant, including both lianescent and tree-shrub species, such as *Quercus ilex* L., *Ficus carica* L., *Rubus ulmifolius* Schott, *Rubia peregrina* L., *Pinus halepensis* Mill., *Pistacia lentiscus* L., *Celtis australis* L., and invasive species *Ailanthus altissima* (Mill.) Swingle. Other abundant chamaephytes include *Helichrysum italicum* (Roth) G. Don, and *Satureja montana* L., along with the hemicryptophyte *Dittrichia viscosa* (L.) Greuter. The limestone sourced from the Roman quarry in Pula has been petrologically classified as coquina limestone, a sedimentary rock that consists of transported, abraded, and mechanically sorted fragments of various origins, with entire mollusc shells and other bioclasts making up more than 50% of its volume [36,37]. Coquina has a high porosity, meaning there is significant space between shell particles. Moreover, preliminary research indicates that the orientation of the shells influences the tensile strength properties of coquina [38]. The high porosity of coquina limestone facilitates the colonisation of woody species, including trees and shrubs. Less disturbed areas, particularly fortresses more deeply embedded in natural environments, support the development of Mediterranean plant communities, which typically thrive in valley floors, hills, and coastal cliffs around the Adriatic Sea. Evergreen woodlands, primarily dominated by *Pinus halepensis*, form a mosaic of forest patches interspersed with an understory composed of maquis and phrygana species, such as *Quercus ilex* and *Pistacia lentiscus* [39]. In contrast, more altered areas, including fortresses subject to higher levels of anthropogenic disturbance and frequentation, are characterized by the presence of species with high ecological plasticity like *Ficus carica*, pioneer and rapidly colonizing invasive species such as *Ailanthus altissima* and *Rubus ulmifolius* [40]. Then, the high presence of the two chamaephytes *Helichrysum italicum* and *Satureja montana*, is due to an adaptation to intense sun exposure, higher surface temperatures, and water stress typical of stone structures, surviving dry periods thanks to the high heteromorphism of transpiring organs of many mediterranean species [41,42]. Finally, *Dittrichia viscosa* is a ruderal plant species known for its adaptation to extreme environmental conditions [43].

The third cluster comprises sites distributed across both coastal and coastal–inland areas (0–1 km and 10–65 km), with the exception of Andautonia, which is located further inland (65–165 km). This group is characterized by a high abundance of *Parietaria judaica* L., as observed in all other sites, along with a moderate presence of *Cymbalaria muralis* G. Gaertn., B. Mey. & Scherb. However, a notable feature of this cluster is the absence of *Hedera helix* L., which is present in all other surveyed sites. It includes several archaeological sites examined in this study, such as the Roman temple of Nin, Andautonia, and the Pula Amphitheatre, as well as castles of significant cultural and touristic importance. The absence of *H. helix*, together with the low frequency of phanerophytes, could be attributed to effective conservation and maintenance practices applied to these structures. Such measures likely involve the manual removal of herbaceous species, pruning, and the subsequent application of selective herbicides to manage and eradicate woody species, particularly on vertical surfaces [44]. Moreover, *Cymbalaria muralis* is a calcicole species with high germination rates and vigorous growth. It is typically associated with the class *Cymbalario-Parietarietea diffusae*, which includes chasmophytic vegetation growing on wall habitats and is well represented in the phytosociological spectrum, especially within urban ecosystems [45,46]. More specifically, it belongs to the association *Cymbalarietum muralis* Govs 1966, a pioneer plant community specialized in the colonization of vertical wall surfaces [47,48]. These traits confer a high tolerance to disturbance, allowing the species to rapidly recolonize exposed surfaces following cleaning interventions.

To further explore the factors driving the observed patterns, vegetation response was subsequently analysed in relation to the four variables selected for this study (Figure 4).

Average vegetation coverage was first assessed across elevation, distance from the sea, exposure, and conservation status, revealing statistically significant differences only for elevation and conservation status. Consequently, a more detailed analysis of life form distribution was carried out for these two variables (Figure 5). Life forms, as expressions of plant adaptation to environmental conditions, are key descriptors of climatic regimes, human disturbance, and, in the context of monument colonization, potential hazard levels [27]. In this framework, anthropogenic impact, particularly when combined with increasing urbanisation in the areas surrounding some sites, may facilitate the spread of alien species [49]. Indeed, the alien species *Ailanthus altissima* and *Robinia pseudoacacia* were found in several fortresses in Pula, located in close proximity to urban areas. *Parthenocissus quinquefolia*, *Erigeron sumatrensis* and *Euphorbia prostrata* were instead mostly observed at sites with high touristic activity, such as Pula Castle, the Amphitheatre, and the Roman Temple of Nin. Focusing on *Ailanthus altissima*, this species primarily disperses by wind, with water acting as a secondary vector. It also reproduces vigorously through clonal growth and readily resprouts after cutting, which enhances its invasive potential compared to native species [50]. Owing to these traits, *A. altissima* is among the most frequently recorded invasive species in many historical sites. As highlighted by [51], it demonstrates a remarkable ability to establish in ruderal environments and to rapidly colonise sites that are not adequately managed with herbicidal treatments.

The results observed for the elevation can be attributed to the fact that phanerophytes and chamaephytes, typically trees and shrubs, tend to exhibit the highest individual plant cover among life forms. Moreover, they are among the most damaging due to their relatively deep, strong, and perennial root systems, which exert a pronounced mechanical and chemical impact on the stone substrates of walls [4]. This effect is significantly greater when compared to the root systems of other biological types [52,53]. Then therophytes, though generally smaller, can still achieve moderate spatial coverage. Therefore, the predominance of these life forms in the lower elevation ranges results in higher overall vegetation cover. Furthermore, the heterogeneous distribution of life forms contributes to the statistically significant differences observed among elevation ranges. Regarding the state of conservation, an increase in therophytes may be attributed not only to their adaptation to specific edaphic and climatic conditions, but also to the maintenance activities carried out at the sites [52].

Beyond species composition and coverage, diversity patterns were investigated using the Shannon index in relation to the selected variables, revealing no statistically significant variation with respect to exposure (Figure 6). Generally, elevational gradients in species richness can follow different patterns, such as a monotonic decline with increasing elevation or a hump-shaped distribution. These trends are typically driven by a combination of climatic conditions, spatial constraints, evolutionary history, and biotic interactions. However, well-defined patterns are less common along short elevational gradients, as in the present study [54,55]. Nevertheless, the reduction of this index at mid-elevations (150–300 m a.s.l.) possibly reflects suboptimal microclimatic conditions for some species in this range. In contrast, both lower (0–150 m) and higher elevations (>300 m) supported greater floristic diversity, likely due to more stable or favourable environmental conditions.

The results observed for distance from the sea may be attributed to the high environmental heterogeneity and dynamic species composition typically found in coastal areas, which contribute to elevated ecological diversity [56]. Furthermore, they tend to support progressively higher species richness, which in turn enhances floristic diversity [57].

Finally, the analysis of conservation status revealed the highest levels of floristic diversity in sites with low or no maintenance, whereas well-maintained structures exhibited a significant reduction in diversity. This pattern supports the hypothesis that structural degradation in poorly maintained sites increases microhabitat availability by providing more niches and cracks, often resulting from salt crystallization or the expansion of fungal hyphae, which, in turn, create humid microsites that facilitate spontaneous colonization by more vascular plants [58,59]. Conversely, conservation practices such as the application of biocides, manual removal of herbaceous vegetation, mowing, and cutting at the base of trees and shrubs [16,60] can reduce the ecological potential of stone substrates, thereby limiting plant establishment and overall species diversity.

As a final step, the potential impact of deteriogenic flora at the study sites was evaluated using the Hazard Index (Figure 7). Statistically significant differences emerged only across elevation and distance ranges, whereas exposure and conservation status showed no significant effects. These findings can be further interpreted by examining the conservation condition of the sites within each distance and elevation range.

Across the first three distance bands, a high proportion of sites lack regular cleaning and maintenance, such as most fortresses in Pula and several castles not formally recognized for their cultural value. Notable exceptions include Grobnik, Benković, Morosini, and Fiume Castles, which are currently well-maintained and frequently host cultural events.

Low HI corresponded to a higher frequency of well-preserved sites with recognized touristic and cultural relevance, such as the archaeological site of Andautonia and Medvedgrad Castle.

As for elevation, the highest HI values were observed in the 150–300 m a.s.l. range, which includes only one site with a good conservation status, Benković Castle. In the remaining elevation bands, particularly in the highest one (300–600 m a.s.l.), a greater proportion of well-maintained sites is present, while abandoned sites are less frequent, contributing to the lower overall HI values.

Despite apparent differences in values, the various conservation statuses did not show statistically significant differences. Therefore, as done for the other variables, the mean Hazard Index was initially calculated and analysed to explore potential differences among conservation classes. However, due to the lack of statistically significant results, the maximum Hazard Index per replicate was also considered to better understand the observed patterns. Both metrics were evaluated using one-way ANOVA followed by Tukey’s post-hoc tests. In both cases, no statistically significant differences emerged among the conservation statuses. These results suggest that, despite observable differences in floristic diversity and total vegetation cover, the overall hazard potential remains comparable across conservation levels. This outcome is likely due to compensatory effects between high- and low-risk species and the high variability observed within each conservation class.

Considering these observations, exposure was the only variable that did not show statistically significant differences in any of the analyses performed. This aligns with previous findings, as exposure is generally not considered among the most relevant ecological factors [61]. Moreover, the absence of a noticeable effect could lie in the physical properties of limestone. As shown in previous studies, rising temperatures can increase cumulative pore volume in calcareous stone, leading, at certain thresholds, to internal chemical reactions that weaken the stone’s microstructure, reduce cohesion, promote cracking, and ultimately lower mechanical resistance [62]. This behaviour, associated with limestone’s intrinsic porosity, may contribute to buffering microclimatic extremes and reducing exposure-related stress across all orientations. Additionally, other studies, such as [17], have suggested that stone surfaces may function as inanimate “nurse” structures, promoting the establishment and survival of vascular plants even under stressful conditions, particularly on sun-exposed surfaces. However, these findings were based on substrates such as tuff, piperno, and plasters, which differ in porosity, thermal mass and water retention capacity. In contrast, the lithological uniformity of the sites investigated in this study may have reduced the influence of exposure on colonisation dynamics.

Nevertheless, the implementation of an effective maintenance plan remains crucial. Such a plan should aim to remove plants as soon as they become established or start growing, through the integration of both manual and chemical control methods [63,64]. However, the complete eradication of deteriogenic organisms is rarely feasible, primarily due to the high economic costs and the logistical difficulties involved in accessing the upper portions of monuments [65]. Moreover, maintenance interventions should also aim to preserve species of conservation concern, including endemic taxa, to ensure a balanced integration of the site’s natural and cultural values [66]. This is particularly relevant, as the preservation of such sites contributes directly to the protection of floristic richness and overall biodiversity [67].

## 4. Materials and Methods

### 4.1. Study Sites

The study was conducted at 40 historical sites across Croatia, including castles, towers, ancient city walls, fortresses, and archaeological sites, covering Dalmatia, Istria, and various inland areas (Figure 8). Some of the surveyed castles are part of the historical heritage of the prominent Croatian noble family, the Frankopan, whose estates were primarily located in the Gorski Kotar region and on the island of Krk (Figure 9).

The study sites are located across a diverse altitudinal range, from sea level up to 600 m a.s.l., and cover varying distances from the coast, extending from 0 to 165 km inland. To further contextualize the environmental variability across the study area, a map illustrating Croatia’s climatic zones (a) and relative precipitation conditions for the year 2024 (b) is provided (Figure 10).

This variability made it possible to examine how environmental factors, such as elevation, exposure, distance from the sea, and the conservation status of the artefacts, influence the distribution of plant species. In this context, construction material is not a key factor in explaining variations in plant distribution, as limestone is the predominant material, particularly abundant throughout Croatia. Limestone is a sedimentary rock primarily composed of calcium carbonate (CaCO_3_), and it is generally considered hard and resistant. However, when it contains significant amounts of clay or other impurities, it can become friable, soft, and highly absorbent [70]. While the intrinsic properties of limestone generally limit plant colonisation, we hypothesise that specific mineralogical compositions can increase its susceptibility to biological weathering. The selection of study sites was based on their historical and architectural relevance, with a preference for structures that had not undergone visible modern restoration or material replacement. Although comprehensive restoration documentation was not always available, site selection was based on visual inspection of building materials and architectural features suggestive of historical authenticity to maintain consistency of substrate conditions and ensure that the analysis focused on authentic heritage contexts. For analytical purposes, the distance from the sea was grouped into four distinct categories: 0–1 km (coastal sites), 1–10 km (proximal inland sites), 10–65 km (intermediate inland sites), and 65–165 km (deep inland sites). Similarly, elevation was stratified into four bands: 0–50 m a.s.l. (low-altitude coastal sites), 50–150 m a.s.l. (lower hill zones), 150–300 m a.s.l. (mid-altitude regions), and 300–600 m a.s.l. (higher elevation sites). This classification approach was carefully structured to reflect gradual environmental shifts while ensuring an even distribution of sites across the designated categories, thereby enhancing the assessment of their impact on vegetation patterns.

### 4.2. Data Collection and Analysis

Field surveys were conducted from August to October 2024, covering a total of 452 plots (Table 1), a period that captures peak vegetation cover in Mediterranean environments and facilitates the detection of drought-adapted and perennial species. At each site, surfaces oriented towards the four cardinal directions (north, south, east, and west) were selected for analysis, except in cases where walls were missing or inaccessible. On each selected surface, between one and three randomly placed 1 × 1 m plots were examined to capture variations in plant cover and species composition. Only vertical surfaces were considered to ensure consistency in sampling conditions across all sites.

For each sampling unit, the collected data encompassed the site name, geographic coordinates (WGS 84 system in decimal degrees), substrate type, exposure, distance from the sea, and elevation. Plant cover was estimated using the Braun-Blanquet [71] abundance–dominance scale, which was subsequently converted into percentage values as follows: 5 = 88%; 4 = 63%; 3 = 38%; 2 = 15%; 1 = 5%; + = 1%. To evaluate the potential impact of deteriogenic plant species on structural integrity, the Hazard Index (HI) was applied following the methodology proposed by Signorini [72,73]. This index, which ranges from 1 (low risk) to 10 (high risk), integrates three key factors: life form, invasiveness and vigour, and root system characteristics. For the analysis, the Hazard Index was multiplied by the mean coverage percentage of each species. Plant species were identified directly in the field, with uncertain cases verified at the Department of Agricultural Botany, University of Zagreb Faculty of Agriculture, Croatia. Identification was based on multiple floristic references, including *Flora d’Italia* [74,75], *Flora Europaea* [76,77], *Flora Croatica* [78], and the *Flora Croatica Database* [79]. Taxonomic nomenclature was standardized according to the World Flora Online database [80], while angiosperm families were classified following the APG IV system [81]. Author abbreviations adhered to the guidelines of Brummitt and Powell [82]. Vegetation analyses incorporated both life forms and chorological types to explore their distribution patterns and relationships with environmental parameters. Life forms were categorized following Raunkiær’s classification [83] and confirmed through field observations, whereas chorological types were assigned according to [75,84,85]. To assess plant community composition across the study sites, a cluster analysis was conducted using the Bray–Curtis similarity index. The species contributing most to community differentiation were identified through their association index, with a focus on the 40 most frequently occurring species across sites. Statistical analyses, including one-way ANOVA, were performed using SPSS software (Version 29.0.1.0; IBM Corp., Armonk, NY, USA), with significance set at *p* < 0.05 to evaluate the effects of environmental variables. Post-hoc comparisons were conducted using Tukey’s test.

## 5. Conclusions

By integrating floristic, ecological, and structural data, this study advances our understanding of how vascular plants contribute to the degradation of historical structures in Croatia. Through a multifactorial approach, key environmental and anthropogenic drivers, particularly elevation and conservation status, were identified as major factors influencing vegetation cover and floristic diversity. The analysis of plant life forms further clarified the ecological strategies involved and their relationship with site maintenance, highlighting the adaptive responses of species under varying conservation conditions. Although exposure did not significantly influence colonization dynamics, the observed variation along altitudinal and coastal–inland gradients highlights the importance of more targeted maintenance strategies. The use of the Hazard Index revealed that elevation and distance from the sea significantly influence the presence of high-risk species. Notably, the elevated Hazard Index observed in the 150–300 m elevation range suggests that sites within this band may require more urgent and prioritized conservation interventions than those located at lower or higher elevations. These results reinforce the need to incorporate botanical data into preventive conservation planning, especially in regions where plant-induced damage remains understudied. Unlike other studies conducted on different substrates, all sites in this research were built using limestone, allowing for a focused interpretation of species’ behavior in relation to a single lithotype. Future studies should focus on long-term monitoring of vegetation dynamics and species-specific responses to limestone characteristics and climate fluctuations. The installation of microclimate sensors to monitor surface moisture could offer valuable insights into the interactions between stone properties and plant colonization. A multidisciplinary and inclusive approach, engaging conservation professionals, local authorities, and communities, will be key to ensuring the sustainable preservation of Croatia’s rich architectural heritage.

## Figures and Tables

**Figure 1 plants-14-01773-f001:**
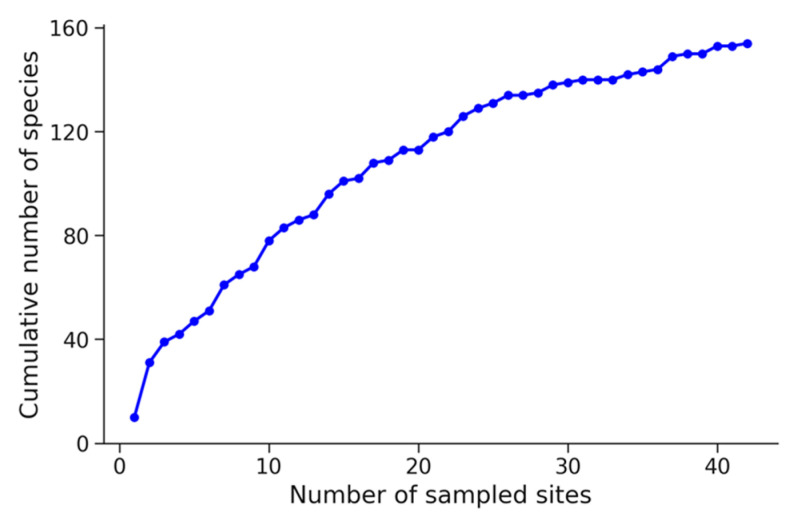
Species accumulation curve as a function of sampling effort. The curve shows the cumulative number of species observed as the number of sampling units increases.

**Figure 2 plants-14-01773-f002:**
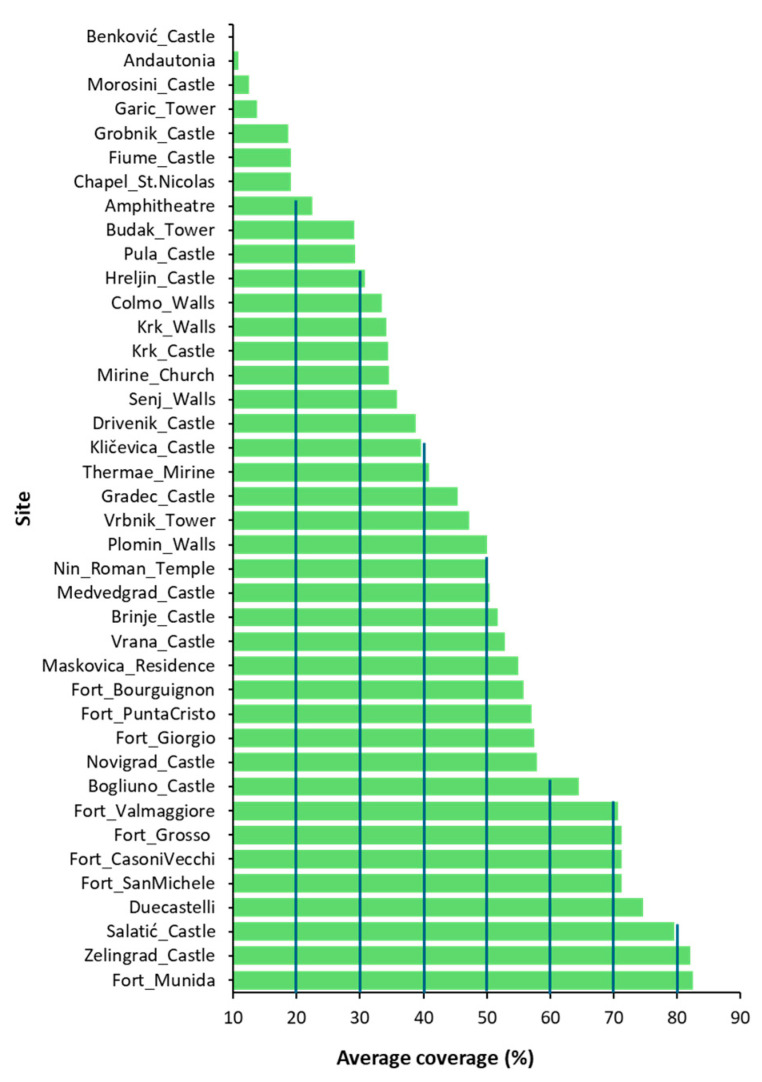
Average coverage across the 40 study sites.

**Figure 3 plants-14-01773-f003:**
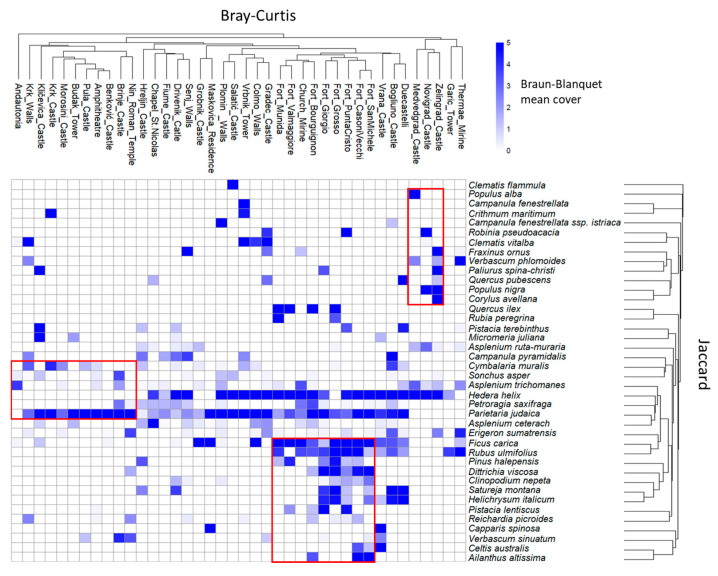
Heatmap with the relative abundance of the 40 most frequent taxa across study sites. Hierarchical clustering of sites is based on the Bray–Curtis dissimilarity, while species are ordered according to the Association Index. The red boxes indicate the three representative clusters.

**Figure 4 plants-14-01773-f004:**
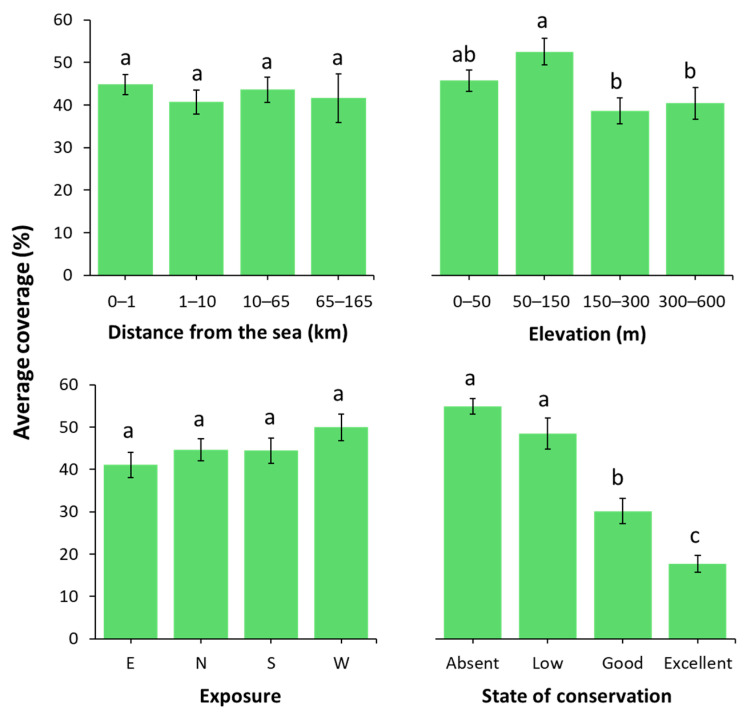
Vegetation coverage across the different variables considered. Error bars represent the standard error of the mean (SEM). Bars sharing the same letter are not significantly different according to the Tukey test (*p* < 0.05).

**Figure 5 plants-14-01773-f005:**
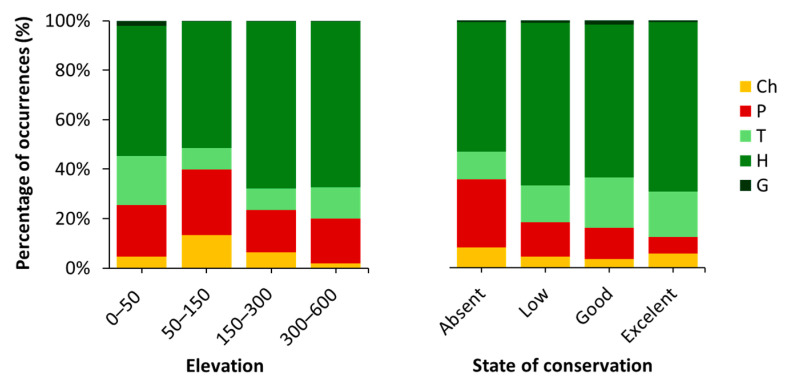
Percentage of species occurrences categorized by life forms across elevation and state of conservation (Ch = Chamaephytes; P = Phanerophytes; T = Therophytes; H = Hemicryptophytes; G = Geophytes).

**Figure 6 plants-14-01773-f006:**
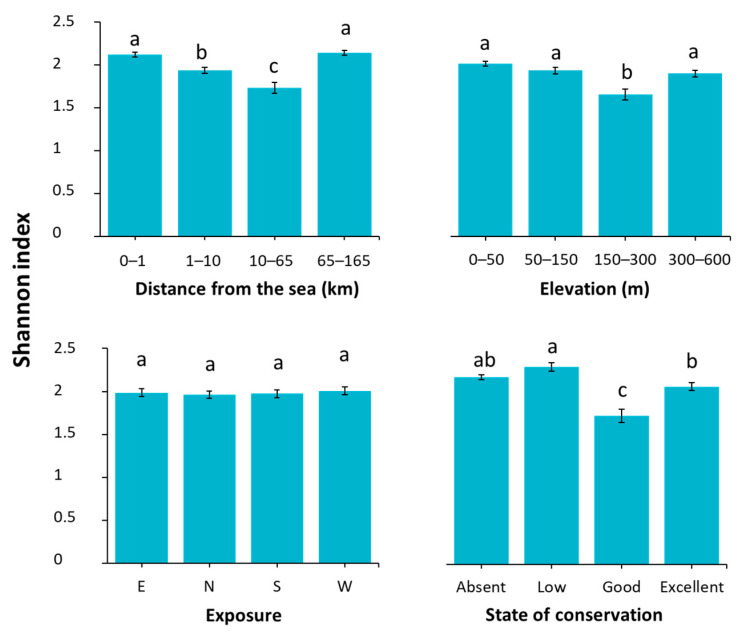
Shannon Index across the different variables considered. Error bars represent the standard error of the mean (SEM). Bars sharing the same letter are not significantly different according to the Tukey test (*p* < 0.05).

**Figure 7 plants-14-01773-f007:**
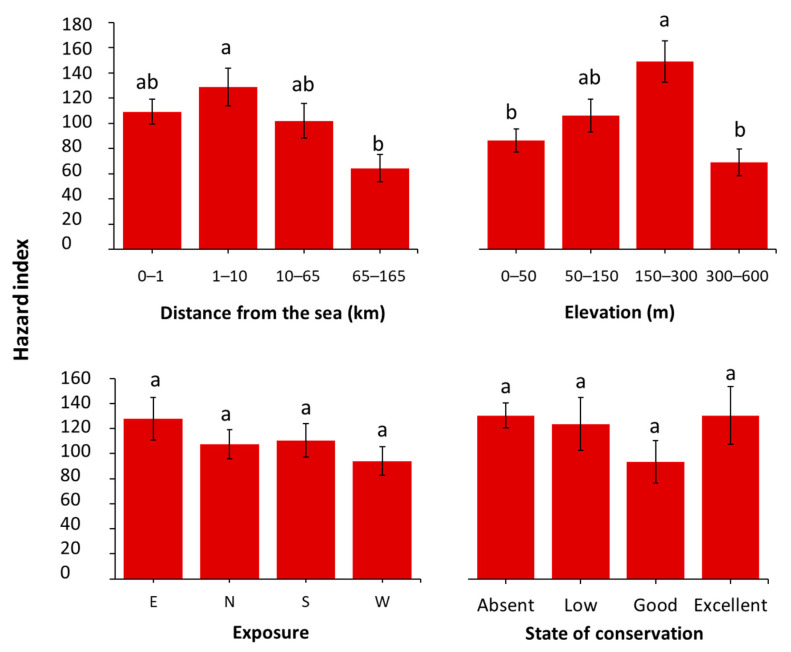
The average Hazard Index (HI) across the different variables considered. Error bars represent the standard error of the mean (SEM). Bars sharing the same letter are not significantly different according to the Tukey test (*p* < 0.05).

**Figure 8 plants-14-01773-f008:**
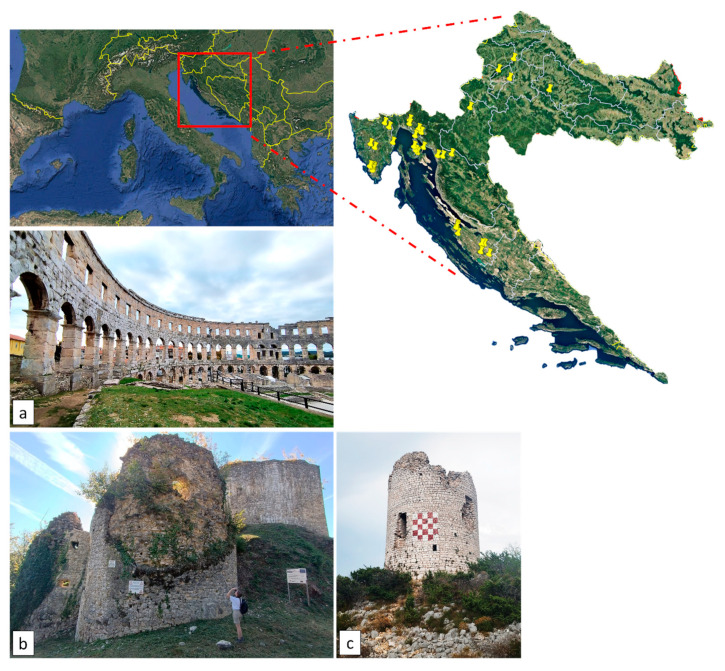
Some of the study sites: (**a**) Amphitheatre of Pula (Istria); (**b**) Zelingrad castle (Kladešćica, inland); (**c**) Budak tower (Dalmatia). The yellow points on the map above represent the surveyed study sites (Photo: authors Alessia Cozzolino and Ivana Vitasović-Kosić, 2024).

**Figure 9 plants-14-01773-f009:**
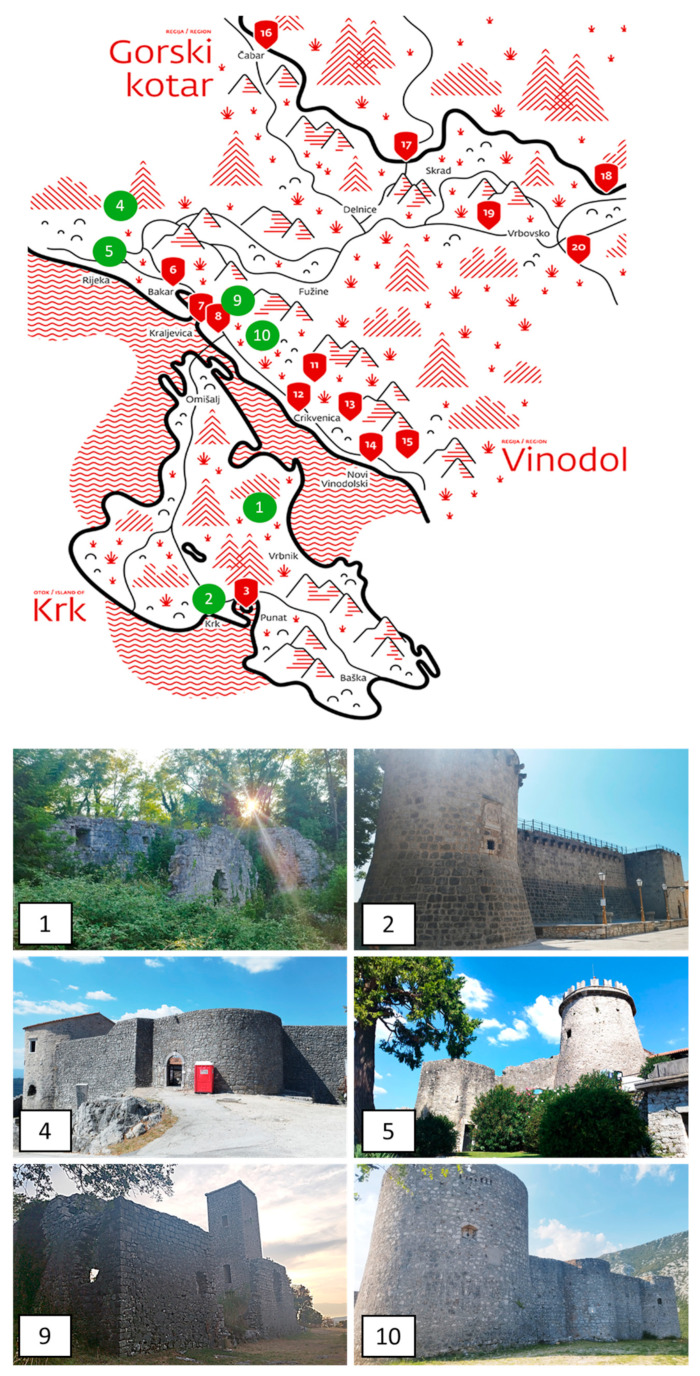
Routes of the Frankopan Castles indicated by numbers from 1 to 20 on the map above. The numbers of the surveyed castles are highlighted in green: (1) Gradec; (2) Krk; (4) Grobnik; (5) Trsat (Fiume); (9) Hreljin; (10) Drivenik. Regional names are displayed in Croatian (Photo: author Alessia Cozzolino, 2024).

**Figure 10 plants-14-01773-f010:**
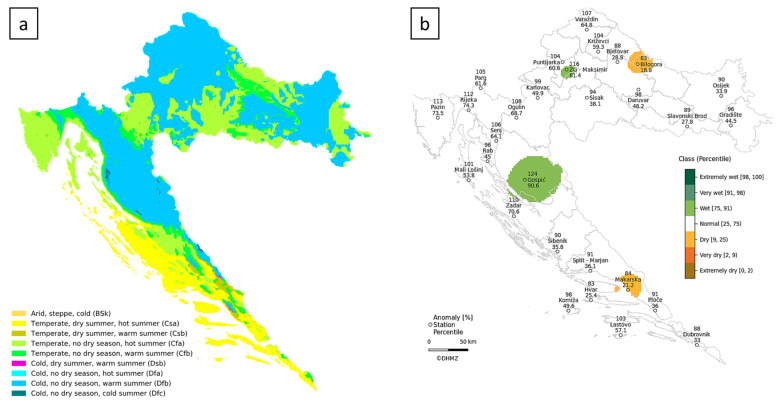
Environmental context of the study area: (**a**) Köppen–Geiger climate classification map for Croatia [68]; (**b**) Precipitation percentiles for the year 2024 relative to the 1991–2020 climatological normal, based on data from the Croatian Meteorological and Hydrological Service (DHMZ), showing how rainfall compares to long-term climatic patterns across the country [69].

**Table 1 plants-14-01773-t001:** Details of the study sites, including their typology and the number of sampling plots. The table also provides geographic coordinates (latitude, longitude) in the WGS 84 reference system.

N.	Study Sites	Type	N. of Plots	Coordinates (WGS 84)
1	Church Fulfinium Mirine (island Krk)	Archaeological site	12	45.20386, 14.54395
2	Mirinski češalj (Thermae Mirine, island Krk)	Archaeological site	12	45.20539, 14.54362
3	Nin Roman Temple	Archaeological site	9	44.24364, 15.18398
4	Frankopanski Kaštel (Town Krk)	Castle	8	45.0257, 14.57641
5	Pula Amphitheatre	Archaeological site	12	44.87323, 13.85024
6	Town Senj	Ancient city walls	12	44.99131, 14.90279
7	Town Krk walls	Ancient city walls	6	45.02839, 14.57579
8	Maškovića Han (Vrana)	Castle	9	43.95412, 15.5489
9	Vrana	Castle	12	43.9557, 15.54961
10	Vrbnik (Krk)	Tower	9	45.0776, 14.6759
11	Pulski Kaštel	Castle	12	44.87022, 13.84549
12	Fort SanMichele (Pula)	Fortress	12	44.8662, 13.8538
13	Fort PuntaCristo (Pula)	Fortress	12	44.8922, 13.79762
14	Fort Valmaggiore (Pula)	Fortress	7	44.88361, 13.80775
15	Fort Bourguignon (Pula)	Fortress	12	44.84683, 13.8335
16	Fort CasoniVecchi (Pula)	Fortress	12	44.85268, 13.84598
17	Fort Giorgio (Pula)	Fortress	12	44.8819, 13.85554
18	Fort Munida (Pula)	Fortress	10	44.88203, 13.81502
19	Fort Grosso (Pula)	Fortress	12	44.88646, 13.81028
20	Salatić (island Krk)	Castle	12	45.04904, 14.54879
21	Andautonia (Zagreb)	Archaeological site	12	45.77357, 16.11724
22	Trsat (Rijeka)	Castle	12	45.33252, 14.4554
23	Dvigrad (Duecastelli)	Castle	15	45.12708, 13.81173
24	Kličevica (Benkovac)	Castle	15	44.03357, 15.56764
25	Drivenik	Castle	12	45.238, 14.6467
26	Plomin	Ancient city walls	12	45.1378, 14.1788
27	Kapela sv. Nikole (Nin)	Archaeological site	12	44.23091, 15.17853
28	Novigrad na Dobri	Castle	12	45.48236, 15.45296
29	Gradec (island Krk)	Castle	14	45.09989, 14.62847
30	Benković (Benkovac)	Castle	12	44.03389, 15.6097
31	Budak (Stankovci)	Tower	12	43.92441, 15.68013
32	Hreljin	Castle	12	45.27466, 14.60233
33	Boljun (Bogliuno)	Castle	12	45.30163, 14.12177
34	Morosini (Svetvinčenat)	Castle	8	45.08814, 13.88235
35	Hum (Colmo)	Ancient city walls	10	45.34828, 14.05054
36	Zelingrad	Castle	12	45.98154, 16.19656
37	Garić grad (Podgarić)	Tower	8	45.63118, 16.75696
38	Grobnik	Castle	12	45.37133, 14.4606
39	Brinje	Castle	12	44.99834, 15.13162
40	Medvedgrad (Zagreb)	Castle	12	45.86909, 15.94023
	Tot		452	

## Data Availability

Data are available upon request.

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
