# Peer review of "Vascular Flora on Croatian Historic Structures: Drivers of Biodeterioration and Conservation Implications"

_plants, 2025, doi:10.3390/plants14121773_

Round 1

Reviewer 1 Report

Comments and Suggestions for Authors

The paper deals with the interesting topic of the biodeterioration due to higher plants on monuments, based on the analysis of several cases of Croatian monuments. The paper is original and analyses for the first time such area. However, a major revision, with modification and improvement of the paper, is needed, in order to clarify some weak points, or not sufficiently clarified questions. A better check of the literature is also needed.

In detail:

The introduction lacks punctuation and some sentences should be separated, since in the present form the structure is not well made. The evaluation of references should be better done.

Some of them (e.g. 2, 3) don’t fit at the best- please add and change). On the contrary I suggest improving the literature, in the different parts, and using the suggested literature for discussion.

 When considering Biodeterioration, I suggest citing and commenting the results of:

Hosseini, Z. and Caneva, G., 2021. Evaluating hazard conditions of plant colonization in Pasargadae World Heritage Site (Iran) as a tool of biodeterioration assessment. International Biodeterioration & Biodegradation, 160, p.105216.

Gunasdi, Y., Aksakal, O. and Kemaloglu, L., 2023. Biodeterioration of some historical monuments in Erzurum by vascular plants. International Biodeterioration & Biodegradation176, p.105530.

 Similarly, when considering Ecology of biodeterioration and nature conservation, I suggest citing and commenting:

Caneva, G., Pacini, A., Grapow, L.C. and Ceschin, S., 2003. The Colosseum's use and state of abandonment as analysed through its flora. International Biodeterioration & Biodegradation, 51(3), pp.211-219.

Kumbaric, A., S. Ceschin, Vincenzo Zuccarello, and G. Caneva. "Main ecological parameters affecting the colonization of higher plants in the biodeterioration of stone embankments of Lungotevere (Rome)." International Biodeterioration & Biodegradation 72 (2012): 31-41.

Panitsa, M., Tsakiri, M., Kampiti, D. and Skotadi, M., 2024. Archaeological Areas as Habitat Islands: Plant Diversity of Epidaurus UNESCO World Heritage Site (Greece). Diversity16(7), p.403.

 Later, I suggest separating Results and Discussions

 In the results:

A Table showing the distribution of species in the 40 monuments is really welcomed and useful for further studies (you can use values of frequency to avoid numbering 452 plots, such as in the synoptic phytosociological tables).

 Data of Figure 2, which presents the average species coverage across the study sites, is really surprising ( a cover of 70-80 % of plants on stone monuments is really high!... Please give some comments…..).

 Some of the listed plants are endemic or subendemic, can you do some comment about their significance and the importance of maintaining such habitats (See for example Panitsa and other already cited papers...e.g. Zangari et al...).

 The observation that “distance from the exposure do not show statistically significant differences, suggesting that these environmental factors are not primary drivers of vegetation cover” can be reinforced considering some previous papers (see for example Kumbaric et al. 2012) .

 Methodology:

I am also surprised in the possibility of analysing carefully 40 monuments with 452 plots in only 3 months! Do you have previous data?

A more detailed description of stone materials is needed to obtain correlations among such parameter and plant colonization.

The parameter of inclination is also highly relevant, please give some comment of it.

More importantly, the use of Signorini index was recently improved by other A. that you cite (See Caneva et al…..), why don’t use such method? It is more complex, but it gives more interesting results.

Author Response

Reviewer #1.

Comm: The paper deals with the interesting topic of the biodeterioration due to higher plants on monuments, based on the analysis of several cases of Croatian monuments. The paper is original and analyses for the first time such area. However, a major revision, with modification and improvement of the paper, is needed, in order to clarify some weak points, or not sufficiently clarified questions. A better check of the literature is also needed.

In detail:

The introduction lacks punctuation and some sentences should be separated, since in the present form the structure is not well made. The evaluation of references should be better done.

Some of them (e.g. 2, 3) don’t fit at the best- please add and change).

Res: >We would like to thank the reviewer for his evaluable comments.

>Regarding the introduction, new references have been added, and some sentences have been modified as >follows:

  • Heritage plays a crucial role in fostering community cohesion and identity, serving as a fundamental pillar for social and community harmony [2]”. Lines 37-39.
  • These factors are further compounded by biological agents, including bacteria, lichens, algae, fungi, bryophytes, and vascular plants that cause several damages [4]”. Lines 41-42.
  • Consequently, the resulting alterations can be both chemical and physical. Biochemical damage occurs through direct metabolic activity with production of acids, chelating substances, alkalis, enzymes, and pigments. In contrast, biophysical damage is associated with mechanical degradation processes, such as the penetration and expansion of organisms within the material, or through cyclical hydration and dehydration [6]”. Lines 44-49.

Comm: On the contrary I suggest improving the literature, in the different parts, and using the suggested literature for discussion.

When considering Biodeterioration, I suggest citing and commenting the results of:

Hosseini, Z. and Caneva, G., 2021. Evaluating hazard conditions of plant colonization in Pasargadae World Heritage Site (Iran) as a tool of biodeterioration assessment. International Biodeterioration & Biodegradation160, p.105216.

>Done

Gunasdi, Y., Aksakal, O. and Kemaloglu, L., 2023. Biodeterioration of some historical monuments in Erzurum by vascular plants. International Biodeterioration & Biodegradation176, p.105530. >Done

 Similarly, when considering Ecology of biodeterioration and nature conservation, I suggest citing and commenting:

Caneva, G., Pacini, A., Grapow, L.C. and Ceschin, S., 2003. The Colosseum's use and state of abandonment as analysed through its flora. International Biodeterioration & Biodegradation51(3), pp.211-219. >Done

Kumbaric, A., S. Ceschin, Vincenzo Zuccarello, and G. Caneva. "Main ecological parameters affecting the colonization of higher plants in the biodeterioration of stone embankments of Lungotevere (Rome)." International Biodeterioration & Biodegradation 72 (2012): 31-41. >Done

Panitsa, M., Tsakiri, M., Kampiti, D. and Skotadi, M., 2024. Archaeological Areas as Habitat Islands: Plant Diversity of Epidaurus UNESCO World Heritage Site (Greece). Diversity16(7), p.403. >Done

Later, I suggest separating Results and Discussions

>Done

Comm: In the results:

A Table showing the distribution of species in the 40 monuments is really welcomed and useful for further studies (you can use values of frequency to avoid numbering 452 plots, such as in the synoptic phytosociological tables).

>Done. A new table A2 has been added to the supplementary material.

Comm: Data of Figure 2, which presents the average species coverage across the study sites, is really surprising ( a cover of 70-80 % of plants on stone monuments is really high!... Please give some comments…..).

>Done. To explain these values, additional argumentation has been added as follows: “An analysis of average coverage across the 40 study sites (Figure 2) reveals substantial variability, largely influenced by their geographic location and the extent of maintenance activities, which are often associated with their touristic significance. Focusing on sites with over 50% vegetation coverage, these areas are typically located within densely wooded environments, where the absence of regular cleaning and maintenance fosters the proliferation of arboreal species. Notably, this includes the fortresses of Pula and the castles of Zelingrad, Salatić, and Duecastelli, where extensive growth of lianescent climbers such as Hedera and Clematis was also observed.”. Lines 212-219.

Comm: Some of the listed plants are endemic or subendemic, can you do some comment about their significance and the importance of maintaining such habitats (See for example Panitsa and other already cited papers...e.g. Zangari et al...).

Res. >Done. To highlight the presence of endemic taxa, some relevant results on chorological form have been added in the Results section, lines 108-114: “From a chorological perspective, several alien taxa were identified during the surveys, including Ailanthus altissima (Mill.) Swingle, Robinia pseudoacacia L., Parthenocissus quinquefolia (L.) Planch., Erigeron sumatrensis Retz., Erigeron annuus (L.) Desf., Euphorbia prostrata Aiton, and Euphorbia maculata L. In contrast, the endemic taxa recorded were Campanula fenestrellata subsp. fenestrellata, Campanula fenestrellata subsp. istriaca (Feer) Damboldt, Centaurea dalmatica A. Kern., and Centaurea spinosociliata Seenus”. Then, a sentence was added in the Discussion as follows: “Moreover, maintenance interventions should also aim to preserve species of conservation concern, including endemic taxa, to ensure a balanced integration of the site's natural and cultural values [66]. This is particularly relevant, as the preservation of such sites contributes directly to the protection of floristic richness and overall biodiversity [67].”. Lines 417-421.

Comm:  The observation that “distance from the exposure do not show statistically significant differences, suggesting that these environmental factors are not primary drivers of vegetation cover” can be reinforced considering some previous papers (see for example Kumbaric et al. 2012).

Res: >Done. A sentence has been added in the Discussion as follows: “Considering these observations, exposure was the only variable that did not show statistically significant differences in any of the analyses performed. This aligns with previous findings, as exposure is generally not considered among the most relevant ecological factors [61]”. Lines 396-399.

 Comm:Methodology:

I am also surprised in the possibility of analysing carefully 40 monuments with 452 plots in only 3 months! Do you have previous data?

Res: We appreciate the reviewer's concern regarding the feasibility of surveying 40 monuments and 452 plots in a three-month period. Most of the sites surveyed had a limited spatial extent, which made it possible to efficiently complete floristic surveys at several locations within the same day. The sampling design was optimised to balance the needs of spatial coverage and data quality. Moreover, although the number of surveys may appear remarkably high, it should be noted that they were conducted within a limited timeframe during one of the authors’ research stays abroad as part of her PhD project. Consequently, the available time was carefully optimized by maintaining an intense and tightly scheduled work pace.

Comm: A more detailed description of stone materials is needed to obtain correlations among such parameter and plant colonization.

Res: >Done. We added some details regarding limestone material as follows: “Limestone is a sedimentary rock primarily composed of calcium carbonate (CaCO₃), and it is generally considered hard and resistant. However, when it contains significant amounts of clay or other impurities, it can become friable, soft, and highly absorbent [70]. While the intrinsic properties of limestone generally limit plant colonisation, we hypothesise that specific mineralogical compositions can increase its susceptibility to biological weathering”. Lines 450-455.

Colmm: The parameter of inclination is also highly relevant, please give some comment of it.

Res: >Done. Information on the analysed inclination has been added as follows: “Only vertical surfaces were considered to ensure consistency in sampling conditions across all sites”. Lines 477-478.

Comm: More importantly, the use of Signorini index was recently improved by other A. that you cite (See Caneva et al…..), why don’t use such method? It is more complex, but it gives more interesting results.

RES: >We acknowledge the existence of more recent and comprehensive risk assessment methodologies, which we greatly appreciate for their scientific robustness. The decision to adopt the previously used index in this study was motivated by the intention to maintain methodological consistency with a prior survey conducted on historical sites in Italy, thus allowing for a meaningful comparison between the two datasets. Moreover, it is worth noting that, at the time of the Italian data collection (which began in April 2023), the newly proposed index had not yet been published. Nonetheless, we regard this work as a first step and a useful baseline for future studies, where the integration of more advanced risk models will certainly be considered, particularly for selected sites of conservation concern.

Reviewer 2 Report

Comments and Suggestions for Authors

The manuscript “Plant-Induced Biodeterioration in Croatian Historic Structures: Patterns and Drivers” describes in-depth research on the colonization by vascular plants of 40 sites of cultural heritage significance in Croatia, including castles, towers, and archaeological remains. In the work, after having identified all the colonizing species, the incidence of colonization and coverage of the various species is analyzed and some parameters that could affect the development of the different plants in the analyzed sites are evaluated.

Finally, the Shannon index and, in particular, the Hazard Index are evaluated. This latter index is particularly relevant as it can provide indications on the management of the artistic heritage and its state of conservation.

The final aim of the work can be identified in the development of predictive models that integrate different variables and that could improve the early identification of sites at risk and guide more effective intervention strategies.

The work is well written and comprehensively frames and analyses the problem.

Author Response

Comm: The manuscript “Plant-Induced Biodeterioration in Croatian Historic Structures: Patterns and Drivers” describes in-depth research on the colonization by vascular plants of 40 sites of cultural heritage significance in Croatia, including castles, towers, and archaeological remains. In the work, after having identified all the colonizing species, the incidence of colonization and coverage of the various species is analyzed and some parameters that could affect the development of the different plants in the analyzed sites are evaluated.

Finally, the Shannon index and, in particular, the Hazard Index are evaluated. This latter index is particularly relevant as it can provide indications on the management of the artistic heritage and its state of conservation.

The final aim of the work can be identified in the development of predictive models that integrate different variables and that could improve the early identification of sites at risk and guide more effective intervention strategies.

The work is well written and comprehensively frames and analyses the problem.

Res: >We would like to thank the reviewer for the positive evaluation of our manuscript and for the appreciation of our work.

Reviewer 3 Report

Comments and Suggestions for Authors

see the attachment.

Author Response

Reviewer #3

  1. General Comments

Comm: This manuscript makes a valuable contribution to understanding plant-driven biodeterioration in Croatia, addressing a critical gap in cultural heritage conservation. The integration of ecological indices (Shannon Index, Hazard Index) and multivariate analyses strengthens the scientific rigor of the study. However, key weaknesses include methodological ambiguities (e.g., site selection criteria, Hazard Index calculation, and insufficient contextualization of results within the broader literature. The reliance on non-academic sources (e.g., Wikipedia) and unresolved chronological inconsistencies (e.g., data collection in 2024 for a 2025 publication) further undermine credibility. Given these issues, a major Revision is recomended to enhance clarity, methodological transparency, and academic rigor.

Res: >We are grateful to the reviewer for the valuable feedback and suggestions.

  1. Specific Comments

2.1 the Title

Comm: For more clarity, I suggest revising the title to read: "Deteriogenic Vascular Flora in Croatian Historic Structures: Drivers of Biodeterioration and Conservation Implications," as this aligns better with the appropriate terminology and emphasizes the applied outcomes.

RES: >Done. The revised title more accurately captures the focus of the study. A slight modification was made to avoid repetition of the terms “deteriogenic” and “biodeterioration”: Vascular Flora on Croatian Historic Structures: Drivers of Biodeterioration and Conservation Implications.

 2.2 the Keywords

COMM: I think the term "higher plants" used on page 1, line 29 feels outdated, and I suggest we replace it with "vascular plants" for better precision. Additionally, the phrase "Croatian monuments" seems redundant given the title, so I feel it might be worthwhile to omit that as well. I believe we should focus on concepts like vascular plants, cultural heritage conservation, Raunkiær life forms, limestone bioreceptivity, and invasive species to enhance clarity and relevance.

RES: >Done. We appreciate the reviewer’s suggestions. We changed some key words with “invasive species, vascular plants, cultural heritage conservation, Raunkiær life forms”.

 2.3 The Abstract

COMM: I think the abstract needs some revisions. I feel there are some key findings that are missing, particularly the Hazard Index results. The phrase "elevation and conservation status" is repeated, but it doesn't quantify the impact of these factors. I suggest specifying numerical results, such as mentioning that "vegetation cover decreased by 13–16% in well-conserved sites (p < 0.05)" and that "mid-elevation sites (150–300 m) exhibited the highest Hazard Index (e.g, HI = 160)."

RES:>Done. We added relevant numerical results in the Abstract as follows: “As expected, vegetation cover decreased significantly, by 67.75%, from sites classified as 'Absent' to those with 'Excellent' conservation status (p < 0,001)”. Lines 20-21.

The highest risk levels were recorded at mid-elevations (150–300 m), where the Hazard Index reached its maximum value (HI = 158)”. Lines 26-28.

2.4 The Introduction

COMM: The introduction needs a clearer gap statement. It notes that Croatian studies are "understudied" (Line 19), but adding examples of relevant work from nearby areas, like Mediterranean Europe, could help provide context. Additionally, more explanation regarding the focus on limestone substrates is necessary; for instance, comparing Croatia’s biodeterioration research with that of countries like Italy or Greece might offer useful insights. Citing works by Motti & Bonanomi (2018) could support this. It’s important to clarify why the findings on limestone matter, as its prevalence in Croatian heritage sites contrasts with a limited understanding of its bioreceptivity compared to materials like sandstone or granite. This difference is significant. Lastly, it would be beneficial to add a paragraph about invasive species, such as Ailanthus altissima, to highlight a local threat, especially since "ecological" is mentioned.

RES: >Done. To reinforce the concept of a lack of biodeterioration studies in Croatia, we have added examples as follows: “In neighbouring countries such as Italy and Greece, numerous historic and archaeological sites have been studied for biodeterioration, yielding important insights and contributing to their scientific and cultural valorisation [15,20,24,26]. In contrast, this phenomenon remains poorly investigated in Croatian monuments. A comprehensive study in this context could offer valuable contributions to the understanding of Croatia’s cultural heritage, fostering an integrated perspective between botany and history”. Lines 84-91.

Regarding the focus on limestone, a sentence has been added in the Introduction: “All surveyed structures are built exclusively with limestone, a material whose physical and chemical properties influence its bioreceptivity and response to plant colonisation”. Lines 79-81.

A paragraph interpreting the results has been added to the Discussion section (lines 206–212): “Floristic distribution in this study was assessed in relation to environmental and management variables. In many studies, including that of Motti et al. [17], substrate is considered an additional factor influencing floristic composition, particularly when multiple substrate types are present, as this allows for correlation analyses between vegetation and lithological variability. In the present case, however, all sites were built exclusively on limestone; for this reason, substrate was not included among the variables considered.”.  

Other information has been added in lines 450-455: “Limestone is a sedimentary rock primarily composed of calcium carbonate (CaCO₃), and it is generally considered hard and resistant. However, when it contains significant amounts of clay or other impurities, it can become friable, soft, and highly absorbent [70]. These properties generally limit plant colonization, although certain mineralogical compositions can increase limestone's susceptibility to biological weathering”.

In accordance with the reviewer’s suggestion, a sentence about invasive species has been added in the Introduction section: “In particular, the analysis revealed the presence of both endemic and invasive vascular plant species, highlighting the need to consider floristic composition not only from a conservation biology perspective but also in relation to management priorities”. Lines 91-94.

Then, a paragraph interpreting the results has been added to the Discussion section: “In this framework, anthropogenic impact, particularly when combined with increasing urbanisation in the areas surrounding some sites, may facilitate the spread of alien species [49]. Indeed, the alien species Ailanthus altissima and Robinia pseudoacacia were found in several fortresses in Pula, located in close proximity to urban areas. Parthenocissus quinquefolia, Erigeron sumatrensis and Euphorbia prostrata were instead mostly observed at sites with high touristic activity, such as Pula Castle, the Amphitheatre, and the Roman Temple of Nin. Focusing on Ailanthus altissima, this species primarily disperses by wind, with water acting as a secondary vector. It also reproduces vigorously through clonal growth and readily resprouts after cutting, which enhances its invasive potential compared to native species [50]. Owing to these traits, A. altissima is among the most frequently recorded invasive species in many historical sites. As highlighted by [51], it demonstrates a remarkable ability to establish in ruderal environments and to rapidly colonise sites that are not adequately managed with herbicidal treatments.”. Lines 315-327.

2.5 The Materials and Methods

2.5.1 Study Site Description

COMM: The site selection criteria provided on page 12, come across as somewhat vague. I feel that while the exclusion of "modernized/restored sites" is mentioned, it remains undefined, which could lead to confusion. Additionally, although geographic distribution like Dalmatia, Istria, and inland areas is noted, I think it overlooks the important aspect of microclimatic variability—such as the differences between coastal and continental climates. To improve this, I suggest defining what is meant by "modernization/restoration," perhaps including a specification for sites that have been treated with biocides within the past five years. I also believe it would be helpful to include a map (Figure 8) illustrating climate zones or rainfall gradients to better contextualize the environmental variability.

RES: We sincerely apologize for the confusion. Sites were selected based on the absence of modern restoration interventions, with priority given to castles, towers, archaeological sites, and historic city walls that visibly retained their original building materials. While full documentation of restoration history was not always available, structures showing no clear evidence of recent treatments, such as synthetic mortars or material replacement, were preferred. This criterion aimed to ensure that vegetation patterns could be observed on surfaces as close as possible to their original state, preserving consistency in substrate conditions across sites. To be clearer, we corrected the relative sentence as follows: “The selection of study sites was based on their historical and architectural relevance, with a preference for structures that had not undergone visible modern restoration or material replacement. Although comprehensive restoration documentation was not always available, site selection was based on visual inspection of building materials and architectural features suggestive of historical authenticity to maintain consistency of substrate conditions and ensure that the analysis focused on authentic heritage contexts”. Lines 455-461.

To better contextualize the environmental variability, a map illustrating climate zones and rainfall gradients has been added as figure 10: “To further contextualize the environmental variability across the study area, a map illustrating Croatia’s climatic zones (a) and relative precipitation conditions for the year 2024 (b) is provided (Figure 10)”. Lines 437-440.

2.5.2 Data collection and analysis

COMM: I think the data collection period mentioned for "August to October 2024" conflicts with the manuscript’s planned publication date in 2025. It seems to me that sampling during a single season could introduce some bias, particularly favoring drought-tolerant species. I suggest clarifying the timeline to something like "August–October 2022". Additionally, I feel that justifying the single-season sampling could strengthen the arguments, for example, noting that "Late summer surveys capture peak vegetation cover in Mediterranean environments.”

RES: >Done. Additional information has been added to the materials and methods section as follows: “Field surveys were conducted from August to October 2024, covering a total of 452 plots (Table 1), a period that captures peak vegetation cover in Mediterranean environments and facilitates the detection of drought-adapted and perennial species”. Lines 471-473.

Although conducting the surveys within a single season may introduce a degree of bias, it is important to note that the fieldwork was carried out during one of the author's research stays abroad, as part of a PhD project that had only recently begun in 2022. Given the limited timeframe, the available time was carefully optimized through a rigorous and tightly scheduled work plan.

COMM: The conversion of the Braun-Blanquet scale to percentages, as mentioned on page 13, lines 352-353, seems to lack proper validation. I think that the use of arbitrary conversions, such as the notation of "+ = 1%', could potentially misrepresent the cover effectively. I feel like it would be more beneficial to refer to standardized protocols, like the one proposed by van der Maarel in 1979, or at least provide a clear rationale for the conversions that have been chosen. This would help in ensuring that the data is communicated accurately and transparently.

RES: We appreciate the reviewer’s comment regarding the conversion of Braun-Blanquet cover values into percentage estimates. In our study, we adopted a widely used approximation of the Braun-Blanquet scale, originally proposed in 1946. This method has been extensively applied in vegetation studies, particularly in the Mediterranean region, and slightly modified in several recent works, which we followed as a reference in our approach:

  • Bonanomi, G., Mingo, A., Incerti, G., Mazzoleni, S., & Allegrezza, M. (2012). Fairy rings caused by a killer fungus foster plant diversity in species‐rich grassland. Journal of Vegetation Science23(2), 236-248.
  • Motti, R., Bonanomi, G., & Stinca, A. (2020). Deteriogenic flora of the Phlegraean Fields Archaeological Park: ecological analysis and management guidelines. Nordic Journal of Botany38(5).
  • Bonanomi, G., Iacomino, G., Di Costanzo, L., Moreno, M., Tesei, G., Allegrezza, M., ... & Idbella, M. (2025). Mechanisms and impacts of Agaricus urinascens fairy rings on plant diversity and microbial communities in a montane Mediterranean grassland. FEMS Microbiology Ecology101(4).

COMM: I think the Hazard Index (HI) calculation, as outlined on Page 13, Lines 354–356, isn’t explained thoroughly enough. The method referenced from Signorini 1996 feels quite outdated, and I feel that the risk factors such as life form, invasiveness, and root system don’t have sufficient justification for their weighting. I suggest we provide a clearer detail of the HI formula, potentially stating it as: "HI = (Life Form Score × Invasiveness Score × Root System Score) × % Cover." Additionally, it might be beneficial to reference more modern frameworks, like the RHV, to enhance the relevance of our approach.

RES: The reviewer’s comment raises an important point, which agrees with a similar observation made by another reviewer. We acknowledge the existence of more recent and comprehensive risk assessment methodologies, which we greatly appreciate for their scientific robustness. However, considering the scope of this study, which involved numerous sites distributed across a wide geographical area, and the limited timeframe available, we opted for a simplified and previously validated method that allowed for rapid and consistent data collection. We also note that the application of improved index was not feasible in this context due to the lack of certain specific input data required by those approaches. Nonetheless, we regard this work as a first step and a useful baseline for future studies, where the integration of more advanced risk models will certainly be considered, particularly for selected sites of conservation concern.

2.6 The Results and Discussion Section

COMM: The figure interpretations seem a bit too text-heavy and lack the necessary statistical nuance. For instance, the claim of "significant differences" on Page 6, Line 198 could be strengthened by also reporting the F- and p-values. Additionally, while the Bray-Curtis dendrogram on Page 3, Figure 3 is described, I feel it would be helpful to link it to the environmental drivers that are influencing the results. I think it would enhance clarity if ANOVA results were reported, such as stating, "Elevation influenced cover (F(3,36) = 5.2, p = 0.004)". It would also be beneficial to relate the clusters presented in Figure 3 to the site characteristics outlined in Table 1; for example, mentioning that "Cluster 1 sites were inland with higher forest coverage" would provide valuable context.

RES: >Done. The F and p values have been added as follows (Figure 4): “Distance from the sea and exposure do not show statistically significant differences, suggesting that these environmental factors are not primary drivers of vegetation cover (F(3,386) = 0.416, p = 0.742; and F(3,440) = 1.551, p = 0.201, respectively). In contrast, elevation and conservation status show significant differences among subgroups (F(3,402) = 4.259, p = 0.006; and F(3,448) = 37.261, p < 0.001, respectively).”. Line 149-153.

The F and p values have been added as follows (Figure 6): “Diversity was assessed using the Shannon Index, which revealed no significant differences in relation to exposure (F(3,440) = 0.164, p = 0.92), in contrast to the other variables considered: elevation (F(3, 448) = 14.679, p < 0.001), distance from the sea (F(3, 448) = 19.620, p < 0.001), and conservation status (F(3, 448) = 20.573, p < 0.001)”. Lines 169-172.

The F and p values have been added as follows (Figure 7): “The analysis revealed statistically significant differences only across altitudinal and distance ranges (F(3,448) = 6.678, p < 0.001; and F(3,448) = 2.80, p = 0.039, respectively), while no significant effects were found for exposure and conservation status (F(3,440) = 1.551, p = 0.201, and F(3,448) = 0.975, p = 0.404, respectively)”. Lines 190-193.

For Figure 3, ANOVA results comparing the three clusters in relation to environmental variables, namely, elevation and distance from the sea, have been added: “One-way ANOVA tests revealed significant differences among the three floristic clusters identified in the heatmap, both in terms of elevation (F(2, 19) = 7.178, p = 0.005) and distance from the sea (F(2, 19) = 16.691, p < 0.001). Post hoc comparisons using Tukey’s HSD test showed that Cluster 1 included sites at significantly higher elevations (mean = 361 m a.s.l.) and greater distances from the sea (mean = 113,152 m) compared to Cluster 2 (mean elevation = 43 m, mean distance = 396 m), while Cluster 3 occupied intermediate positions in both variables and did not differ significantly from the other clusters”. Lines 128-134.

Lastly, to relate the clusters presented in Figure 3 to the site characteristics outlined in Table 1, several sentences fave been added as follows: “Notably, the study sites belonging to this cluster are located in inland regions of Croatia (65-165 km), where natural woodland vegetation is more prevalent”. Lines 233-235; “The second cluster includes sites located in the coastal area (0-1 km), particularly the Church of Mirine and, predominantly, the fortresses in Pula, constructed by the Austro-Hungarian Empire to protect the strategic naval port [33]”. Lines 254-256; “The third cluster comprises sites distributed across both coastal and coastal-inland areas (0-1 km and 10-65 km), with the exception of Andautonia, which is located further inland (65-165 km)”. Lines 287-289.

COMM:  The discussion seems to overlook some conflicting literature that I think is important to acknowledge. For example, the null effect of exposure mentioned on page 9, line 288, contrasts sharply with studies that link solar exposure to biofilm formation. While invasive species impacts, like those from Robinia pseudoacacia, are noted, I feel like they could be better understood if compared to global datasets. It might also be helpful to discuss why exposure had no noticeable effect; for instance, limestone’s high porosity may buffer microclimatic extremes, reducing exposure-related stress. Additionally, I think it would enrich the analysis to compare Habitat Integrity (HI) values to similar indices, for instance, the risk scores outlined by Motti & Bonanomi in 2018 for Italian castles. Lastly, addressing Ailanthus altissima’s role in biodeterioration in relation to other Mediterranean species could provide deeper insights into this issue.

RES: We thank the reviewer for these comments.

We agree that in studies focusing on microbial colonisation, such as biofilm formation, solar exposure often plays a critical role due to its influence on moisture retention and phototrophic activity. However, in research concerning vascular plant colonisation, exposure does not always emerge as a significant driver of vegetation variability. As reported by Kumbaric et al. (2012), exposure may have a limited or inconsistent effect, particularly when other factors (e.g. elevation or disturbance) are more influential. We added a sentence to the Discussion section to acknowledge this point and clarify the context of our findings: “This aligns with previous findings, as exposure is generally not considered among the most relevant ecological factors [61]”. Lines 397-399.

Moreover, the absence of significant differences in hazard patterns across exposures in our dataset could lie in the physical properties of limestone. In this context a paragraph to better contextualise the result has been included in the Discussion section, with reference also to the study of Motti and Bonanomi (reference 17): “Moreover, the absence of a noticeable effect could lie in the physical properties of limestone. As shown in previous studies, rising temperatures can increase cumulative pore volume in calcareous stone, leading, at certain thresholds, to internal chemical reactions that weaken the stone’s microstructure, reduce cohesion, promote cracking, and ultimately lower mechanical resistance [62]. This behaviour, associated with limestone’s intrinsic porosity, may contribute to buffering microclimatic extremes and reducing exposure-related stress across all orientations. Additionally, other studies, such as [17], have suggested that stone surfaces may function as inanimate “nurse” structures, promoting the establishment and survival of vascular plants even under stressful conditions, particularly on sun-exposed surfaces. However, these findings were based on substrates such as tuff, piperno, and plasters, which differ in porosity, thermal mass and water retention capacity. In contrast, the lithological uniformity of the sites investigated in this study may have reduced the influence of exposure on colonisation dynamics”. Lines 399-411.

Regarding the role of Ailanthus altissima in biodeterioration compared to other Mediterranean species, its impact can be explained by its efficient dispersal mechanisms and its ability to outcompete native vegetation. A new sentence has been added in the paragraph related to invasive species, as follows: “Focusing on Ailanthus altissima, this species primarily disperses by wind, with water acting as a secondary vector. It also reproduces vigorously through clonal growth and readily resprouts after cutting, which enhances its invasive potential compared to native species [50]”. Lines 321-324.

Finally, the impact of Robinia pseudoacacia has been argued in lines 250-253: “… and Robinia pseudoacacia, an invasive non-native tree. This invasive non-native tree is a pioneer species that rapidly colonizes poor, disturbed, or degraded environments, as well as forest edges, such as those found in the castle’s surroundings [31,32]”.

2.8 The Conclusion

COMM: I feel like the conclusion is a bit too general, particularly with phrases like "tailored management strategies" mentioned in line 374, which seem to lack actionable steps. I think it would be more effective if we specified some strategies. For instance, we could prioritize mid-elevation sites, specifically those between 150 and 300 meters, for invasive species removal. Additionally, installing microclimate sensors to monitor limestone moisture could provide valuable insights.

RES: >Done. We appreciate the reviewer’s suggestion. Two sentences have been added as follows: “Notably, the elevated Hazard Index observed in the 150–300 m elevation range suggests that sites within this band may require more urgent and prioritized conservation interventions than those located at lower or higher elevations. These results reinforce the need to incorporate botanical data into preventive conservation planning, especially in regions where plant-induced damage remains understudied”. Lines 519-524; “The installation of microclimate sensors to monitor surface moisture could offer valuable insights into the interactions between stone properties and plant colonization”. Lines 528-530.

2.9 The References

COMM: I noticed that non-academic sources, such as "Pula, Wikipedia" cited on Line 150, could be an issue for the credibility of the work. I think it would strengthen the paper to replace Wikipedia with peer-reviewed sources, such as Girardi-Jurkic 2012, which discusses Pula’s limestone.

RES: >Done. The previous citation has been replaced with a more relevant reference: Vojnović, N. The Potential of Memorial Tourism Development in Istria (Croatia). Hrvatski geografski glasnik/Croatian Geographical Bulletin 2021, 82, 107–129, doi:10.21861/HGG.2020.82.02.04.

2.10 The Language Issues

COMM: I noticed some minor language issues, such as inconsistencies in capitalization, like "Frankopan Castles" versus "Frankopanski Kaštel." Additionally, some sentences feel a bit too lengthy, which can really reduce readability. For instance, instead of saying, "It is precisely these properties that make this type of limestone highly bioreceptive..." I think it would be clearer to simplify it to something like, "Coquina limestone’s high porosity facilitates colonization by shrubs." I suggest we use "deteriogenic" throughout for consistency. To avoid such language issues, plz thoroughly proofread the entire MS.

RES: >Done. We would like to clarify that the term 'Frankopanski Kaštel' refers to a specific castle, whereas 'Frankopan' is the surname of the noble family mentioned in the manuscript. Accordingly, the expression 'Frankopan Castles' refers collectively to all the castles associated with this family, including those featured in the routes illustrated in Figure 9.

We thank the reviewer for his suggestions. We modified the sentence as follows: “The high porosity of coquina limestone facilitates the colonisation of woody species, including trees and shrubs”. Lines 271-272.

Round 2

Reviewer 1 Report

Comments and Suggestions for Authors

The paper has been improved according to the suggestions in most parts, excluding the Harzard evaluation, but it can be accepted in the present form

Reviewer 3 Report

Comments and Suggestions for Authors

 I appreciate the authors' dedication to improving the quality of the manuscript. They addressed all the comments/concerns that I raised. So I don't have any further comments, and suggest the manuscript be accepted in its current form.